# Association between Habitual Tea Consumption and Metabolic Syndrome and Its Components among Chinese Adults Aged 18~59 Years: Based on China Nutrition and Health Surveillance 2015–2017

**DOI:** 10.3390/nu14173502

**Published:** 2022-08-25

**Authors:** Yuxiang Yang, Dongmei Yu, Wei Piao, Kun Huang, Liyun Zhao

**Affiliations:** NHC Key Laboratory of Trace Element Nutrition, National Institute for Nutrition and Health, Chinese Center for Disease Control and Prevention, Beijing 100050, China

**Keywords:** tea consumption, metabolic syndrome, Chinese adult, nutrition surveillance

## Abstract

Background: Tea consumption is widely reported to have beneficial effects on metabolic functions. The current study is to evaluate the association between habitual tea consumption and risk for metabolic syndrome and its components among Chinese adults aged 18~59 years. Methods: 43,757 participants aged 18~59 years from China Nutrition and Health Surveillance 2015–2017 were included and divided into four groups based on the amount of daily tea consumption in the current study. Using multiple-adjustment logistic regression to explore the relationship between habitual tea consumption and metabolic syndrome-related health outcomes. Results: Compared with those who did not consume tea habitually, participants who drank over 5 cups of tea per day showed a significantly lower risk of metabolic syndrome (OR = 0.836, 95% CI = 0.771–0.905), blood pressure elevated (OR = 0.906, 95% CI = 0.845–0.972), triglyceride elevated (OR = 0.797, 95% CI = 0.741–0.857), and fasting plasma glucose elevated (OR = 0.772, 95% CI = 0.715–0.833), but higher risk for central obesity (OR = 1.354, 95% CI = 1.236–1.484). Regardless of gender, higher tea consumption was related to lower risk of triglyceride and fasting blood glucose elevated but higher risk for central obesity. While for protective effect on metabolic syndrome, blood pressure elevated, and HDL-C reduction only showed in females. Conclusions: Results from current study support that habitual tea consumption would benefit metabolic syndrome and its related components, especially among females.

## 1. Introduction

Metabolic syndrome (MetS) is a series of abnormal metabolic disorders mainly represented by abdominal obesity, insulin resistance, hypertension, and dyslipidemia [1]. Mets has association with several adverse health outcomes, including cardiovascular disease (CVDs), diabetes, cancers, renal diseases, cognitive impairment, all-cause mortality, etc. [2,3,4]. According to previous studies, about one third of Chinese adults are affected by MetS currently, which would lay great burden on both individual wellbeing and health care system [5]. Thus, there is an urgent need for exploring population-based strategies related to MetS prevention and control, which would further benefit for the health condition of whole life span [2].

Diet and lifestyle habits play a key role on the prevention and management of MetS [6], and among them, tea consumption is an important part of dietary habits, which has also been widely reported to have beneficial effects on Mets in epidemiological and clinical research [7]. Tea (mainly types including green, black, and oolong) is one of the most widely consumed beverages worldwide, and rich in several minerals, vitamins, amino acid, caffeine, and polyphenols [8,9]. Due to its ability of anti-inflammation and antioxidant, tea has the prospective to be a nonpharmacological strategy to manage several non-communicable chronic diseases (NCDs), such as hypertension, hyperglycemia, dyslipidemia, and obesity, which are also the components of MetS [10,11]. Several systematic reviews and meta-analyses had shown that tea consumption would lower the likelihood of having MetS [9,12]; however, they still lack evidence on tea consumption and the risk of MetS among nationwide Chinese adults, especially for observational studies. Therefore, based on the data from China Nutrition and Health Surveillance (CNHS), a national representative survey in 2015–2017, our current study thoroughly explored the association between tea consumption and risk of MetS and its components separately among Chinese adults aged 18~59 years, and it may provide a new scope on MetS prevention and control in China.

## 2. Methods

### 2.1. Participants

All the information of the participants was obtained from CNHS 2015–2017 which was conducted among Chinese adults aged over 18 from 31 provinces/municipalities/autonomous in mainland China. To ensure that the nationwide samples were consistent with the national situation in terms of social and economic development and population composition, a stratified, multistage, and random sampling method was applied at the beginning of CNHS 2015–2017. Detailed information about CNHS was illustrated in the previous report [13]. Participants in the current study were selected based on the process shown in Figure 1. All the participants had signed the informed consent before the survey, and this project was approved by the Ethics Committee of the Chinese Center for Disease Control and Prevention (approval number: 201519-B).

### 2.2. Basic Information

In a face-to-face manner, well-trained investigators from the local Center for Disease Control and Prevention (CDC) collected the socioeconomic and lifestyle information of the interviewees, including living area, household income, education, marital status, smoking habits, daily sleeping duration, physical activities, sedentary behaviors, etc. Afterward, for keeping the accuracy and facticity, all the questionnaires were filled by investigators under strict quality control by senior CDC staffs.

### 2.3. Dietary Information

A validated 64-item food frequency questionnaire (FFQ) was used to investigate the food intakes including staple foods, beans, vegetables, fruits, bacteria and algae, milk, meat, aquatic products, eggs, beverages, alcohol, and others in the past 12 months [14]. Daily average consumption was calculated according to the consumption frequency (times/per day, week, month, or year) and weight (in gram) per time. Daily intake of edible oil and condiments per capita was calculated by asking the consumption of edible oil and condiments and the number of people who usually eat at home for each meal (breakfast, lunch, and dinner) within the whole family in the past month. Finally, according to the China Food Composition Tables (2009 and 2018 edition), the average daily energy and nutrient intake of each participant was calculated [15,16].

Meanwhile, a comprehensive review had indicated that the Dietary Approaches to Stop Hypertension (DASH-diet) could serve as a strategy for MetS treatment and management [17]. Thus, we also calculated sex-specific DASH-diet score for each participant in the current study. Due to the low consumption of low-fat dairy products among Chinese population, dairy and related products were used to replace this item [18]. Daily intake of fruits, vegetables, legumes and nuts, whole grains, dairy and related products, sodium, red meat and processed meat, and sugary beverages were categorized by quintiles (scored 1 to 5), of which the first five items are given scores in ascending order, and the last three items in descending order. Finally, the sum of each food score was used for representing the adherence of the DASH diet [19]. Then, the participants were separated into four groups according to the quartile of DASH-diet score.

### 2.4. Medical Examination

Physical measurement and laboratory tests were conducted to obtain the height, weight, waist circumference (WC, cm), blood pressure (BP, mmHg), fasting plasma glucose (FPG, mmol/L), serum total cholesterol (TC, mmol/L), triglycerides (TG, mmol/L), high-density lipoprotein cholesterol (HDL-C, mmol/L), and low-density lipoprotein cholesterol (LDL-C, mmol/L) of the participants. They were measured on an empty stomach in the morning. Corresponding methods of the above medical examination have been described in previous studies [20].

### 2.5. Definition of Tea Consumption

Tea consumption of each participant was collected by asking how many cups of tea they usually drank every day (based on 200 mL per cup), regardless of the type of tea they consumed. Habitual tea drinkers in the current study were deemed to be those who drank one or more cups of tea every day. Then, participants were divided into non-habitual drinkers, 1~2 cups/day, 3~4 cups/day, and more than 5 cups/day.

### 2.6. Definition of MetS

Diagnostic criteria of MetS are based on National Cholesterol Education Program-Adult Treatment Panel III (NCEP-ATP III) shown in Table 1 [21]. And participants who had MetS were diagnosed by reaching three or more components below.

### 2.7. Potential Confounders

To conduct multiple adjustment in the logistic regression model, the socioeconomic covariables were defined as follows: gender (male/female); living area (urban/rural); educational level (primary school or below/junior middle school/high school or above); per capita annual income of household (not given/<10,000 RMB/10,000~<25,000 RMB/≥25,000 RMB); marital status (married/other status). Moreover, family history of hypertension and diabetes was deemed to be having lineal relatives (including grandparents, parents, or siblings) diagnosed.

Covariables which reflect lifestyle habits were categorized as follows: body mass index (BMI, kg/m^2^) was separated into underweight (BMI < 18.5), normal (18.5 ≤ BMI < 24), overweight (24 ≤ BMI < 28), and obese (BMI ≥ 28) [20]; smoking status was separated into current smoker and non-smoker; alcohol drinking status was separated into excessive (≥25 g/day for males and ≥15 g/day for females) or not [21]; physical activity was separated into low, medium, and high calculated from total metabolic equivalent (MET) and duration of different level of physical activities within a week [22]; sedentary behavior was separated into <2 h, 2~3 h, and ≥4 h per day; sleep duration was separated into <7 h, 7~8 h, and ≥9 h per day; and medical examination within one year (yes/no).

### 2.8. Statistical Analyses

All the continuous and categorical variables included in the current study were described by mean ± standard deviation ($\bar{x}$ ± SD) and count (*n*, %), respectively. Stepwise adjustment logistic regression was used to examine the association between tea consumption and risk of MetS and its components, and results were given by odds ratio (OR) and related 95% confidential interval (95% CI). Moreover, except for overall analysis, sex–strata analysis was also conducted in the logistic regression model due to the natural difference in body composition and dietary demands between males and females. Subgroup analysis was conducted by age group, physical activity, sleep duration, smoking, and alcohol drinking status. In addition, *p*-value <0.05 was considered to have statistical significance. All the statistical analyses in the current study were conducted by SAS v9.4 (SAS Institute Inc., Cary, NC, USA).

## 3. Results

### 3.1. Basic Characteristics

A total of 43,757 participants aged 18~59 years from CNHS 2015–2017 were finally included in the current study. Basic characteristics of participants categorized by habitual tea consumption are shown in Table 2. In general, participants who were males, elderly, normal weight, living in rural regions, of lower education level, medium income, and those who have married also tended to have higher tea consumption. Meanwhile, participants who were non-smokers at that time, drank less alcohol, had higher physical activity level, had longer sedentary behaviors, had moderate sleep duration, had poorer DASH-diet adherence, did not take medical examination within one year, and who did not have a family history of hypertension and diabetes tended to consume more tea. Except for physical activity, the composition of other variables was different between groups (*p* < 0.001). For MetS, no statistical significance was observed between groups (*p* = 0.46).

### 3.2. Energy Intake and Biomarkers of MetS

Of different groups of habitual tea consumption, significant increasing trends were observed for daily energy intake, BMI, WC, SBP, DBP, TC, TG, and LDL-C, whereas when taking the means between groups into consideration, the highest means of BMI, TG, and LDL-C were observed in the group which commonly drank 3~4 cups of tea per day. Moreover, no significant linear trend was observed when considering FPG (*p*-trend = 0.063) and HDL-C (*p*-trend = 0.073). Further information is available in Appendix A.

### 3.3. Association between Habitual Tea Consumption and MetS

Results are shown in Table 3. Compared with those who did not consume tea habitually, participants who drank over 5 cups of tea per day showed a significantly lower risk of MetS after multiple adjustment (OR = 0.836, 95% CI = 0.771–0.905, *p*-trend < 0.0001). However, when the participants were analyzed by gender, higher tea consumption showed significant results only in females after multiple adjustment (OR = 0.679, 95% CI = 0.589–0.782, *p*-trend < 0.0001), while in males only a protective effect was observed without statistical significance (OR = 0.975, 95% CI = 0.882–1.078, *p*-trend = 0.7261).

### 3.4. Subgroup Analysis

Results of subgroup analysis by age, physical activity, sleep duration, current smoker, and excessive drinker are shown in Figure 2. Compared with the highest consumption (over 5 cups/day) and the lowest group (none), the protective effect of habitual tea consumption appeared in most of the subgroups. Among them, the protect effect was more pronounced in the elderly (30~<45 years: OR = 0.818, 95% CI = 0.699–0.958; 45~59 years: OR = 0.864, 95% CI = 0.785–0.951), low and moderate sleep duration (<7 h: OR = 0.809, 95% CI = 0.672–0.974; 7~8 h: OR = 0.837, 95% CI = 0.756–0.928), non-smoker (OR = 0.782, 95% CI = 0.704–0.869), and participants who did not drink alcohol excessively (OR = 0.799, 95% CI = 0.731–0.873), while the results were similar among subgroups with different physical activity levels. Moreover, effects modification by current smoking (*p* for interaction = 0.0451) and excessive drinking status (*p* for interaction = 0.0359) were observed. Further information is available in Appendix A.

### 3.5. Association between Habitual Tea Consumption and Each Component of MetS

Compared with those who did not drink tea habitually, participants in the highest tea consumption group showed a higher risk for central obesity (OR = 1.354, 95% CI = 1.236–1.484, *p*-trend < 0.0001) and a lower risk for BP elevated (OR = 0.906, 95% CI = 0.845–0.972, *p*-trend = 0.0002), TG elevated (OR = 0.797, 95% CI = 0.741–0.857, *p*-trend < 0.0001), and FPG elevated (OR = 0.772, 95% CI = 0.715–0.833, *p*-trend < 0.0001) after multiple adjustment, while no significant results was observed as for HDL-C decreased. Stratified by sex, the highest tea consumption was related to higher risk of central obesity, lower risk of TG and FPG elevated among both genders. Moreover, higher tea consumption was associated with lower risk of HDL-C decreased among females, but with slightly higher risk of HDL-C decreased among males. Further information is available in Table 4.

## 4. Discussion

Our current study showed that habitual tea consumption had a protective effect on MetS and some of its components among Chinese samples aged 18~59 years in CNHS 2015–2017. Overall, participants who had a habitual tea consumption of over five cups a day had a lower risk for MetS, BP elevated, TG elevated, and FPG elevated, but a higher risk for central obesity compared with non-drinkers. Stratified by gender, the risk for central obesity, TG, and FPG elevated remained stable with the same trend in overall participants, but the protective effect on MetS and BP elevated only showed in females. Moreover, higher tea consumption also showed a lower risk for HDL-C reduction among female participants. In subgroup analysis, the protective effect on MetS by habitual tea consumption was more pronounced among those who were 30~59 years old, were non-smokers, did not drink alcohol excessively, and slept less than 8 h per day, but only the interaction between smoking and excessive drinking with habitual tea consumption was observed (*p* for interactions <0.05).

To date, the association between tea consumption and MetS and its components has not been well illustrated, and there are contradictions in different results. Our findings are partly consistent with previous studies. Grosso et al. reported that among Polish participants, drinking more tea daily had an inverse association with MetS and central obesity in overall participants [23]. However, after analyzing by gender, the protective effects of MetS only showed in males. In addition, higher tea consumption was also reported to lower the risk for elevated FPG in females. In another study conducted in Poland, no significant result was observed related to tea consumption and MetS, but the results showed that higher tea consumption had inverse association with central obesity among overall and female participants [24]. Vernarelli et al. [25] analyzed tea consumption and markers of MetS among adults aged over 18 years from NHANES, and the results showed that higher tea consumption was inversely associated with BMI and WC for both genders who drank hot tea, but for those who drank iced tea, the results were opposite to the former. As for blood biomarkers, significant results were observed that drinking hot tea could lower FPG among females and higher HDL-C and lower TG among males. However, for those who drank iced tea, higher consumption was related to lower HDL-C and higher TG among females. Meanwhile, it is shown that higher hot tea consumption had an inverse association with C-reactive protein, which was reported to be an indication of a series of metabolic disorders (i.e., hypertension, diabetes, hyperlipidemia, etc.) [26].

Based on previous studies, the healthy benefits against chronic disease as well as MetS of tea consumption may rely on the bioactive substances contained in tea, including polyphenols, caffeine, and some vitamins and minerals [9,27,28,29]. Previous studies have found that tea extract could improve the metabolic status reflected by lipid profile, FPG, weight reduction, etc. [30]. Polyphenols (including epicatechin, catechin, epicatechin gallate, etc.) which are rich in tea, especially in green tea, are reported as the main substances related to the beneficial effects on a series of metabolic disorders due to their antioxidant capacity, which could regulate oxidative stress process and exert anti-inflammatory effects [31,32]. Meanwhile, catechins in green tea were previously reported to be beneficial for regulating central obesity, hypertension, hyperlipidemia, and hyperglycemia, which are the components of MetS [33]. Moreover, as another important substance contained in tea, caffeine also plays a role on antioxidant process and has been hypothesized to have preventive effects on MetS [12,34].

Potential mechanisms related to MetS prevention might be explained as follows: Polyphenols in tea could promote nitric oxide (NO) release and increase its bioavailability and thus inducing vasodilation of arteries [35]. However, consistent with the previous study, the most potent protective effect on blood pressure was observed in the sub-highest tea consumption group instead of the highest one [23]. It could be that caffeine in tea may increase BP due to promoting sympathetic nervous system overactivation and norepinephrine release, which is opposite to polyphenols [12]. It is also reported that consuming tea could enrich probiotics and block the absorption of lipids and proteins in the gastrointestinal tract which could further benefit MetS [7,36,37]. Moreover, green tea could decrease cholesterol in plasma and improve insulin sensitivity to exert its anti-dyslipidemia and anti-hyperglycemia effects by inhibiting or modulating related enzymes and signal pathways [37,38]. Habitual tea consumption was also reported to reduce body fat and WC by inducing thermogenesis and fat oxidation, regulating appetite, and reducing food consumption [39,40]. However, our current research did not find an anti-obesity effect of higher tea consumption, and current results may partly explain that daily energy intake among our participants positively related to habitual tea consumption, which could further lead to overweight and obesity.

Current findings indicated that gender might play an important role in the association between habitual tea consumption and MetS and its components. It could potentially link to the level of sex hormones and related proteins in different genders, such as endogenous sex hormones and sex hormone-binding globulin [41]. Meanwhile, males tended to be smokers and drink more alcohol, which are well-surveyed risk factors for MetS [2], and both reasons might be confounding factors and lead to different biological responses toward tea consumption between males and females. Moreover, interaction effects also showed when stratified by current smoking and excessive drinking status in subgroup analysis; only among those nonsmokers and non-excessive drinkers, a protective association between habitual tea consumption and MetS was observed, indicating that the negative effect of smoking and excessive drinking on MetS might weaken or even cover the positive effect of habitual tea consumption [2].

Our current study has shown that habitual tea consumption might lower the risk for MetS and benefit several components of MetS, such as high blood pressure, hyperglycemia, and hyperlipidemia, etc. In addition, due to economic, trade, and cultural factors, tea is one of the most widely consumed beverages worldwide, and it is estimated that over four million tons of tea are produced every year [37,41]. Thus, finding a new pathway for disease control and prevention through tea consumption is one of the cost-effective as well as nonpharmacological approaches which has broad application prospects [1]. However, application of tea or tea extracts to the prevention of MetS and other NCDs should be carried out with prudence. As previous reports, except for the ability of antioxidant and anti-inflammation, polyphenols in tea could also inhibit the ingesting process of minerals (calcium, magnesium, etc.) which could also lead to some of nutritional problems, though it would not appear under normal tea consumption [7,42]. Moreover, it is reported that using high dosages of green tea extracts might lead to hepatotoxicity and liver damage and interfere with the metabolic process of prescription drugs [43].

To our best knowledge, the current study has the strength of containing a representative adult population in China. We also controlled most of the potential confounders, especially for dietary intake, which is one of the key factors for MetS. Meanwhile, we conducted subgroup analysis to support and robust our current findings. However, some limitations in our study still need to be addressed. Firstly, due to the cross-sectional design, it is hard to draw a causal relationship between habitual tea consumption and risk for MetS and its components. Secondly, we did not collect the types of tea consumed by participants, since different types of tea would contain different kinds of polyphenols, which also indicates different bioactivity and effects deriving from tea [44]. Thirdly, recall bias might exist when collecting information about habitual tea consumption and dietary intakes from participants which might be disparity compared with their authentic condition. Fourthly, conducting such nationwide research demands a considerable variety of resources, and it is hard to control all the potential variables, especially those relying on laboratory tests. Thus, longitudinal research and clinical trials are needed in further research.

## 5. Conclusions

In conclusion, among Chinese adults aged 18~59 years old, higher habitual tea consumption was inversely associated with the risk of MetS and its components (BP elevated, FPG elevated, and TG elevated) but positively with central obesity. After analyzing by gender, the protective effects for FPG and TG elevated but negative effect for central obesity was still significant in both genders, while for MetS, BP elevated, and HDL-C reduction, the protective effects were only observed among female participants. These findings support the potential of habitual tea consumption for preventing MetS and some of its components and further preventing MetS-related diseases. In addition, longitudinal research and clinical trials are still needed to further confirm the health beneficial effects for MetS of tea consumption.

## Figures and Tables

**Figure 1 nutrients-14-03502-f001:**
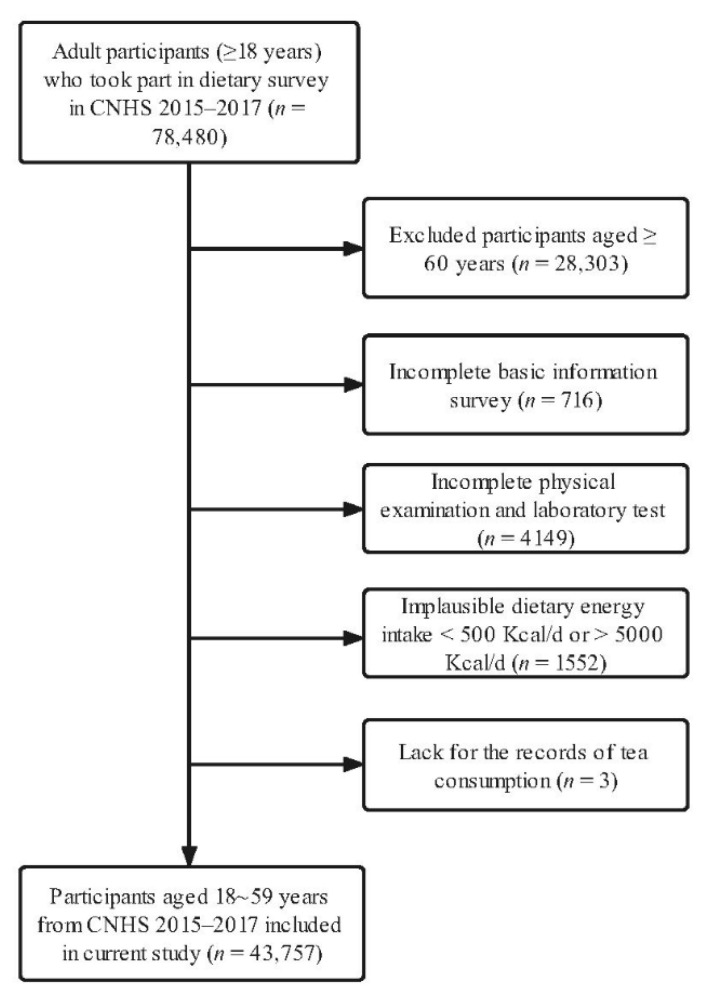
Enrolling process of participants from CNHS 2015–2017.

**Figure 2 nutrients-14-03502-f002:**
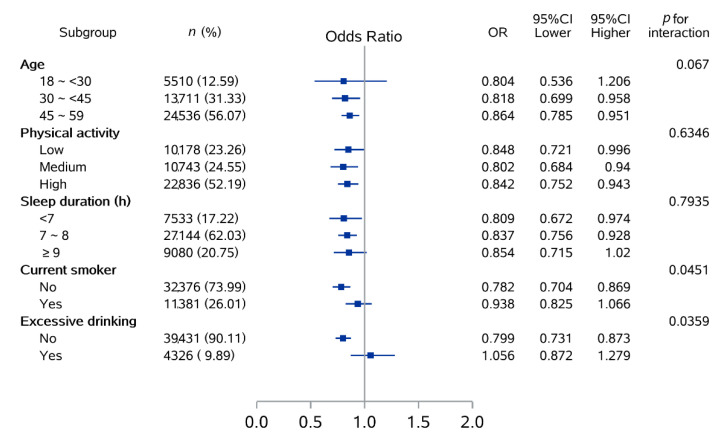
Subgroup analysis between habitual tea consumption and MetS according to potential risk factors.

**Table 1 nutrients-14-03502-t001:** Diagnostic standard of each component of MetS according to NCEP-ATP III.

Components of MetS	Gender	
	Male	Female
WC (cm)	WC ≥ 90 cm	WC ≥ 80 cm
BP (mmHg)	SBP ≥ 130 mmHg and (or) DBP ≥ 85 mmHg and (or) receiving anti-hypertension treatment	
TG (mmol/L)	TG ≥ 1.7 mmol/L and (or) receiving corresponding treatment	
HDL-C (mmol/L)	HDL-C < 1.03 mmol/L and (or) receiving corresponding treatment	HDL-C < 1.3 mmol/L and (or) receiving corresponding treatment
FPG (mmol/L)	FPG ≥ 5.6 mmol/L and (or) receiving anti-diabetes treatment and (or) had pre-diagnosed with diabetes	

**Table 2 nutrients-14-03502-t002:** Basic characteristics of participants aged 18~59 years in CNHS 2015–2017.

Characteristics	Habitual Tea Consumption (*n*, %)			
	None	1~2 Cups/Day	3~4 Cups/Day	Over 5 Cups/Day
Gender *				
Male	11,366 (36.9%)	2495 (53.89%)	2142 (62.38%)	3436 (70.25%)
Female	19,436 (63.1%)	2135 (46.11%)	1292 (37.62%)	1455 (29.75%)
Age (year) *				
18~<30	4328 (14.05%)	569 (12.29%)	294 (8.56%)	319 (6.52%)
30~<45	9561 (31.04%)	1608 (34.73%)	1105 (32.18%)	1437 (29.38%)
45~59	16,913 (54.91%)	2453 (52.98%)	2035 (59.26%)	3135 (64.1%)
BMI *				
Underweight	1116 (3.62%)	148 (3.2%)	96 (2.8%)	187 (3.82%)
Normal	14,585 (47.35%)	2130 (46%)	1539 (44.82%)	2207 (45.12%)
Overweight	10,803 (35.07%)	1610 (34.77%)	1216 (35.41%)	1722 (35.21%)
Obese	4298 (13.95%)	742 (16.03%)	583 (16.98%)	775 (15.85%)
Living area *				
Urban	12,163 (39.49%)	1968 (42.51%)	1443 (42.02%)	2016 (41.22%)
Rural	18,639 (60.51%)	2662 (57.49%)	1991 (57.98%)	2875 (58.78%)
Education level *				
Primary school or below	12,063 (39.16%)	1620 (34.99%)	1240 (36.11%)	1817 (37.15%)
Junior middle school	11,219 (36.42%)	1597 (34.49%)	1246 (36.28%)	1728 (35.33%)
High school or higher	7520 (24.41%)	1413 (30.52%)	948 (27.61%)	1346 (27.52%)
Income *				
Not given	4660 (15.13%)	664 (14.34%)	423 (12.32%)	578 (11.82%)
Low	7420 (24.09%)	1114 (24.06%)	842 (24.52%)	1181 (24.15%)
Medium	11,740 (38.11%)	1655 (35.75%)	1317 (38.35%)	1820 (37.21%)
High	6982 (22.67%)	1197 (25.85%)	852 (24.81%)	1312 (26.82%)
Marital status *				
Married	28,487 (92.48%)	4275 (92.33%)	3227 (93.97%)	4631 (94.68%)
Other status	2315 (7.52%)	355 (7.67%)	207 (6.03%)	260 (5.32%)
Current smoker *				
No	24,503 (79.55%)	3193 (68.96%)	2081 (60.6%)	2599 (53.14%)
Yes	6299 (20.45%)	1437 (31.04%)	1353 (39.4%)	2292 (46.86%)
Excessive drinking *				
No	28,519 (92.59%)	4102 (88.6%)	2929 (85.29%)	3881 (79.35%)
Yes	2283 (7.41%)	528 (11.4%)	505 (14.71%)	1010 (20.65%)
Physical activity				
Low	7139 (23.18%)	1080 (23.33%)	802 (23.35%)	1157 (23.66%)
Medium	7601 (24.68%)	1125 (24.3%)	823 (23.97%)	1194 (24.41%)
High	16,062 (52.15%)	2425 (52.38%)	1809 (52.68%)	2540 (51.93%)
Sedentary behavior (h) *				
<2	3952 (12.83%)	503 (10.86%)	388 (11.3%)	514 (10.51%)
2~3	11,717 (38.04%)	1730 (37.37%)	1301 (37.89%)	1837 (37.56%)
≥4	15,133 (49.13%)	2397 (51.77%)	1745 (50.82%)	2540 (51.93%)
Sleep duration (h) *				
<7	5351 (17.37%)	771 (16.65%)	559 (16.28%)	852 (17.42%)
7~8	18,984 (61.63%)	2958 (63.89%)	2242 (65.29%)	2960 (60.52%)
≥9	6467 (21%)	901 (19.46%)	633 (18.43%)	1079 (22.06%)
DASH-diet *				
Q1	8181 (26.56%)	1203 (25.98%)	974 (28.36%)	1544 (31.57%)
Q2	8229 (26.72%)	1172 (25.31%)	910 (26.5%)	1098 (22.45%)
Q3	7586 (24.63%)	1151 (24.86%)	770 (22.42%)	1057 (21.61%)
Q4	6806 (22.1%)	1104 (23.84%)	780 (22.71%)	1192 (24.37%)
Medical examination *				
No	24,893 (80.82%)	3638 (78.57%)	2714 (79.03%)	3693 (75.51%)
Yes	5909 (19.18%)	992 (21.43%)	720 (20.97%)	1198 (24.49%)
Family history of hypertension *				
No	20,098 (65.25%)	2895 (62.53%)	2176 (63.37%)	2980 (60.93%)
Yes	10,704 (34.75%)	1735 (37.47%)	1258 (36.63%)	1911 (39.07%)
Family history of diabetes *				
No	27,321 (88.7%)	4049 (87.45%)	3011 (87.68%)	4228 (86.44%)
Yes	3481 (11.3%)	581 (12.55%)	423 (12.32%)	663 (13.56%)
MetS				
No	21,366 (69.37%)	3207 (69.27%)	2356 (68.61%)	3434 (70.21%)
Yes	9436 (30.63%)	1423 (30.73%)	1078 (31.39%)	1457 (29.79%)

* Indicated *p* < 0.001; values of polytomous variables may not sum to 100% due to rounding.

**Table 3 nutrients-14-03502-t003:** Association between MetS and habitual tea consumption among overall, male, and female participants.

Habitual Tea Consumption	*n* of Cases (%)	OR (95% CI) *		
		Model I	Model II	Model III
Overall				
None	9436 (30.63)	Ref.	Ref.	Ref.
1~2 cups/day	1423 (30.73)	1.005 (0.94–1.074)	0.993 (0.918–1.074)	0.984 (0.909–1.065)
3~4 cups/day	1078 (31.39)	1.036 (0.96–1.118)	0.925 (0.846–1.012)	0.91 (0.831–0.996)
Over 5 cups/day	1457 (29.79)	0.961 (0.9–1.026)	0.869 (0.803–0.94)	0.836 (0.771–0.905)
*p* for trend	-	0.5415	0.0003	<0.0001
Male				
None	3159 (27.79)	Ref.	Ref.	Ref.
1~2 cups/day	765 (30.66)	1.149 (1.045–1.263)	1.161 (1.04–1.297)	1.137 (1.017–1.27)
3~4 cups/day	653 (30.49)	1.139 (1.03–1.26)	1.037 (0.921–1.168)	0.999 (0.886–1.126)
Over 5 cups/day	1053 (30.65)	1.148 (1.056–1.248)	1.046 (0.948–1.154)	0.975 (0.882–1.078)
*p* for trend	-	0.0002	0.2782	0.7261
Female				
None	6277 (32.3)	Ref.	Ref.	Ref.
1~2 cups/day	658 (30.82)	0.934 (0.848–1.029)	0.866 (0.773–0.97)	0.874 (0.779–0.979)
3~4 cups/day	425 (32.89)	1.028 (0.912–1.159)	0.847 (0.736–0.975)	0.848 (0.737–0.977)
Over 5 cups/day	404 (27.77)	0.806 (0.716–0.907)	0.686 (0.596–0.79)	0.679 (0.589–0.782)
*p* for trend	-	0.0491	<0.0001	<0.0001

* Indicated by odds ratio (OR) and 95% confidence interval (CI). Model I: unadjusted model; Model II: adjusted for gender (in overall analysis), age, BMI; Model III: further adjusted for living area, education level, income, marital status, current smoker (Yes/No), excessive drinking (Yes/No), physical activity, sleeping duration, sedentary behavior, medical examination within one year (Yes/No), DASH-diet score, daily energy intake (Kcal/day), and family history for hypertension and diabetes.

**Table 4 nutrients-14-03502-t004:** Association between different components of MetS and habitual tea consumption among overall, male, and female participants.

Components of MetS	Habitual Tea Consumption	Overall		Male		Female	
		*n* of Cases (%)	OR (95% CI) *	*n* of Cases (%)	OR (95% CI)	*n* of Cases (%)	OR (95% CI)
Central obesity	None	12,948 (42.04)	Ref.	3333 (29.32)	Ref.	9615 (49.47)	Ref.
	1~2 cups/day	1883 (40.67)	1.082 (0.988–1.184)	798 (31.98)	1.184 (1.039–1.35)	1085 (50.82)	0.999 (0.881–1.133)
	3~4 cups/day	1452 (42.28)	1.2 (1.081–1.332)	745 (34.78)	1.317 (1.146–1.513)	707 (54.72)	1.076 (0.918–1.262)
	Over 5 cups/day	1984 (40.56)	1.354 (1.236–1.484)	1201 (34.95)	1.372 (1.221–1.541)	783 (53.81)	1.41 (1.209–1.644)
	*p* for trend	-	<0.0001	–	<0.0001	–	0.0001
BP elevated	None	14,604 (47.41)	Ref.	6151 (54.12)	Ref.	8453 (43.49)	Ref.
	1~2 cups/day	2213 (47.8)	0.929 (0.866–0.996)	1332 (53.39)	0.956 (0.869–1.051)	881 (41.26)	0.912 (0.821–1.012)
	3~4 cups/day	1734 (50.5)	0.889 (0.821–0.963)	1179 (55.04)	0.933 (0.843–1.032)	555 (42.96)	0.854 (0.751–0.973)
	Over 5 cups/day	2608 (53.32)	0.906 (0.845–0.972)	1980 (57.63)	0.952 (0.874–1.036)	628 (43.16)	0.922 (0.815–1.043)
	*p* for trend	-	0.0002	–	0.1389	–	0.0118
TG elevated	None	8491 (27.58)	Ref.	3845 (33.83)	Ref.	4649 (23.92)	Ref.
	1~2 cups/day	1405 (30.35)	1.004 (0.934–1.078)	956 (38.32)	1.185 (1.077–1.303)	449 (21.03)	0.815 (0.727–0.914)
	3~4 cups/day	1055 (30.72)	0.909 (0.837–0.987)	728 (33.99)	0.91 (0.82–1.01)	327 (25.3)	0.976 (0.852–1.118)
	Over 5 cups/day	1440 (29.44)	0.797 (0.741–0.857)	1179 (34.31)	0.913 (0.836–0.996)	261 (17.94)	0.625 (0.541–0.722)
	*p* for trend	-	<0.0001	–	0.0259	–	<0.0001
HDL-C decreased	None	13,093 (42.51)	Ref.	3211 (28.25)	Ref.	9882 (50.84)	Ref.
	1~2 cups/day	1904 (41.12)	1.073 (1.003–1.148)	782 (31.34)	1.133 (1.026–1.251)	1122 (52.55)	1.039 (0.947–1.14)
	3~4 cups/day	1293 (37.65)	0.989 (0.915–1.07)	628 (29.32)	1.025 (0.921–1.142)	665 (51.47)	0.974 (0.867–1.095)
	Over 5 cups/day	1680 (34.35)	0.97 (0.905–1.04)	1021 (29.71)	1.1 (1.005–1.203)	659 (45.29)	0.799 (0.715–0.894)
	*p* for trend	-	0.5893	–	0.0424	–	0.0018
FPG elevated	None	7263 (23.58)	Ref.	3100 (27.27)	Ref.	4163 (21.42)	Ref.
	1~2 cups/day	1076 (23.24)	0.91 (0.843–0.982)	636 (25.49)	0.894 (0.807–0.99)	440 (20.61)	0.94 (0.838–1.055)
	3~4 cups/day	853 (24.84)	0.89 (0.817–0.97)	581 (27.12)	0.897 (0.805–0.999)	272 (21.05)	0.893 (0.773–1.031)
	Over 5 cups/day	1161 (23.74)	0.772 (0.715–0.833)	911 (26.51)	0.808 (0.738–0.885)	250 (17.18)	0.704 (0.608–0.815)
	*p* for trend	-	<0.0001	–	<0.0001	–	<0.0001

* Adjusted for gender (in overall analysis), age, BMI, living area, education level, income, marital status, current smoker (Yes/No), excessive drinking (Yes/No), physical activity, sleeping duration, sedentary behavior, medical examination within one year (Yes/No), DASH-diet score, daily energy intake (Kcal/day), and family history for hypertension and diabetes. Indicated by odds ratio (OR) and 95% confidence interval (CI).

## Data Availability

According to the policy of National Institute for Nutrition and Health, China CDC, data related in this research are not allowed to be disclosed.

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
