# Peer review of "Association between Habitual Tea Consumption and Metabolic Syndrome and Its Components among Chinese Adults Aged 18~59 Years: Based on China Nutrition and Health Surveillance 2015–2017"

_nutrients, 2022, doi:10.3390/nu14173502_

Round 1

Reviewer 1 Report

This is an interesting cross-sectional analysis exploring the association between tea consumption and MetS in the Chinese population. I have a few questions and suggestions below.

1. In table 4, instead of denoting Factor 1, 2, and 3... please name them as the component of Metabolic Syndrome (MetS). This will improve the readability of the results.

2. Author found a positive association between tea consumption and central obesity. Whereas earlier studies (such as Micek et al. and Vernarelli et, al) mentioned by the author report inverse association. I suggest the author discuss these discrepancies as well. Is the sugar consumption along with tea is a reason for a positive association with central obesity? Can sugar intake be adjusted in the model?

3. Lastly, recall bias is an obvious bias that should be listed under the limitations.

Author Response

Point 1: In table 4, instead of denoting Factor 1, 2, and 3... please name them as the component of Metabolic Syndrome (MetS). This will improve the readability of the results.

Response 1: Dear reviewer, thank you for your suggestion. In this revised version, we have updated the information in the Table 4 to make it clearer. Factor 1 to 5 were replaced by the name of each component of MetS, i.e., central obesity, BP elevated, TG elevated, HDL-C decreased, and FPG elevated.

Point 2: Author found a positive association between tea consumption and central obesity. Whereas earlier studies (such as Micek et al. and Vernarelli et, al) mentioned by the author report inverse association. I suggest the author discuss these discrepancies as well. Is the sugar consumption along with tea is a reason for a positive association with central obesity? Can sugar intake be adjusted in the model?

Response 2: Dear reviewer, thank you very much for your reminder. Our current results showed that habitual tea consumption might have positive association with central obesity among Chinese adults aged 18~59 years, which had discrepancies with previous studies[1,2]. We tried to explain this difference by daily energy intake between tea consumption groups in Supplemental Table 1, compared with those who didn’t drink tea habitually, higher consuming group had higher average daily energy intake (Group of None vs. Over 5 cups/day, 2296.3 Kcal/d vs. 2534.2 Kcal/d, p for trend<0.0001), and in the part of discussion, we have discussed this situation in line 78–81 (the part of discussion), “However, our current research didn’t find anti-obesity effect of higher tea consumption, current results may partly explain that daily energy intake among our participants positively related to habitual tea consumption, which could further lead to overweight and obesity.”

Meanwhile, as was shown in Table 2, compared with those who didn’t drink tea habitually, the highest consuming group have higher proportion of current smoker, excessive drinker, and who had longer sedentary behavior, and poorer adherence to DASH-diet, which are potential interfering factors of central obesity. Thus, we have put the above variables into logistic regression model to control confounders.

Moreover, as your suggestion of adjusting sugar intake in the logistic regression model, due to the limitation of the data of CNHS 2015–2017, sugar consumption along with tea was not available, the information of sugar intake could only reflect the sugar consumption for cooking. Thus, we face difficulties on adjusting this variable in our current study, which is one of its limitations. We will try to fix this limitation in our future investigation and make it more accurate.

Point 3: Lastly, recall bias is an obvious bias that should be listed under the limitations.

Response 3: Dear reviewer, thank you for your suggestion. We have corrected this part in the line 116–118 (the part of discussion), “Thirdly, recall bias might exist when collecting information about habitual tea consumption and dietary intakes from participants which might be disparity compared with their authentic condition.” Really appreciate that you have proposed the above suggestions and opinions related to our manuscript, and they have really helped us to improve the whole quality of this study. Thank you very much with our best respects!

Reference

  1. Micek, A.; Grosso, G.; Polak, M.; Kozakiewicz, K.; Tykarski, A.; Puch Walczak, A.; Drygas, W.; Kwasniewska, M.; Pajak, A.; on behalf of, W.I.I.i. Association between tea and coffee consumption and prevalence of metabolic syndrome in Poland - results from the WOBASZ II study (2013-2014). Int J Food Sci Nutr 2018, 69, 358-368, doi:10.1080/09637486.2017.1362690.
  2. Vernarelli, J.A.; Lambert, J.D. Tea consumption is inversely associated with weight status and other markers for metabolic syndrome in US adults. Eur J Nutr 2013, 52, 1039-1048, doi:10.1007/s00394-012-0410-9.

Reviewer 2 Report

The Authors presented a paper about the impact of tea consumption on MetS risk in Chinese population. Althought this topic seems to be interesting for the readers, severeal comments should be addressed.

1. The Authors should explain why did they excluded individuals above 60 years of age?

2. In diagnostic criteria of MetS, the term "fasting plasma glucose" should be used, instead of "fasting blood glucose". Plasma is the reference material to diagnose disoredrs of glucose metabolism.

3. In the "Definiton of tea consumption", the Authors wrote about consumption "regardless of the type of tea they consumed". Ii should be emphasized that different types of tea have different contents of biologically active substances, including antioxidants, therefore they may have different effects on metabolism and health. If data are available on types of tea, the Authors should modify their results.

4. The Authors should explain more clearly the association between higher  tea consumption and obesity.  This relationship seems to be largely dependent on the amount of energy consumed, regardless of the amount of tea consumed. Did the participants declare that they consumed pure tea extract, without any additives such as sugar, syrup, milk, etc., that could raise the overall caloric value and increase the risk of obesity and other metabolic disorders?

Author Response

Point 1: The Authors should explain why did they excluded individuals above 60 years of age?

Response 1: Dear reviewer, thank you for your reminder. One of the reasons is that the previous study based on a longitudinal study among Chinese elderly aged over 60 years old and has reported the association between tea consumption and risk for MetS[1]. Researchers reported that in multiple logistic regression participants who drank tea 5 times per week had a higher risk for the incidence of MetS than non-drinkers (OR=1.38, 95% CI=1.05–1.82, p=0.02), and this association could be still observed among male participants when stratified by gender. However, there is still lack of studies focusing on tea consumption and MetS among Chinese adults in other age groups currently, especially for observational study.

Moreover, another study reported that the prevalence of MetS would increase with aging[2], thus, we consider that aging might play a more significant role in the risk for MetS than habitual consumption. And researchers had pointed out the importance of preventing and treating MetS at younger age[3]. Thus, our current study finally chose the participants aged 18~59 years old as our aim population. And in our further study on this field, we will focus more on the all-aged groups and try to investigate more detailed conditions of tea consumption.

Point 2: In diagnostic criteria of MetS, the term "fasting plasma glucose" should be used, instead of "fasting blood glucose". Plasma is the reference material to diagnose disorders of glucose metabolism.

Response 2: Dear reviewer, thank you very much for your suggestion. We had made this correction in this revised version.

Point 3: In the "Definition of tea consumption", the Authors wrote about consumption "regardless of the type of tea they consumed". It should be emphasized that different types of tea have different contents of biologically active substances, including antioxidants, therefore they may have different effects on metabolism and health. If data are available on types of tea, the Authors should modify their results.

Response 3: Dear reviewer, thank you for your suggestion. As we wrote in method part in line 108-110, “Tea consumption of each participant was collected by asking how many cups of tea they usually drank every day (based on 200ml per cup), regardless of the type of tea they consumed.” Also, in the part of discussion (line 114–116), we had discussed the limitation that “we didn’t collect the types of tea consumed by participants, since different types of tea would contain different kinds of polyphenols, which also indicates different bioactivity and effects deriving from tea[4].” However, due to the limitation of the data in CNHS 2015–2017, only the amount of daily consumption was available, in current study we could only analyze it without the information of the types of tea. But we will try to get more information about tea consumption like types of tea and drinking way, etc. to get more reliable results in future investigations.

Point 4: The Authors should explain more clearly the association between higher tea consumption and obesity. This relationship seems to be largely dependent on the amount of energy consumed, regardless of the amount of tea consumed. Did the participants declare that they consumed pure tea extract, without any additives such as sugar, syrup, milk, etc., that could raise the overall caloric value and increase the risk of obesity and other metabolic disorders?

Response 4: Dear reviewer, thank you for your suggestion. In the main text, we tried to explain this difference by Table 2 and Supplemental Table 1, compared with those who didn’t drink tea habitually, higher consuming group had higher average daily energy intake (Group of None vs. Over 5 cups/day, 2296.3 Kcal/d vs. 2534.2 Kcal/d, p for trend<0.0001), meanwhile, the highest consuming group had higher proportion of current smoker, excessive drinker, and who had longer sedentary behavior, and poorer adherence to DASH-diet, which are potential interfering factors of central obesity and we have put the above variables into logistic regression model to try to get more reliable results.

Also, we have discussed this situation in line 78–81 (the part of discussion), “However, our current research didn’t find anti-obesity effect of higher tea consumption, current results may partly explain that daily energy intake among our participants positively related to habitual tea consumption, which could further lead to overweight and obesity.”

Moreover, as your suggestion, due to the limitation of the data of CNHS 2015–2017, the information of participants who consumed pure tea extract, used any additives such as sugar, syrup, milk, etc. was not available. Thus, we feel hard to adjust these variables in our current study, which is one of the limitations of current study. We will try to fix this limitation in our future investigation and make it more accurate. We really appreciate your efforts for our article, and we attach great importance to your suggestions on our manuscript. The above suggestions did help us improve the quality of our work. Thank you very much with our best respects!

Reference

  1. Dong, X.X.; Wang, R.R.; Liu, J.Y.; Ma, Q.H.; Pan, C.W. Habitual tea consumption and 5-year incident metabolic syndrome among older adults: a community-based cohort study. BMC Geriatr 2021, 21, 728, doi:10.1186/s12877-021-02707-8.
  2. Lechleitner, M. Obesity and the metabolic syndrome in the elderly--a mini-review. Gerontology 2008, 54, 253-259, doi:10.1159/000161734.
  3. Denys, K.; Cankurtaran, M.; Janssens, W.; Petrovic, M. Metabolic syndrome in the elderly: an overview of the evidence. Acta Clin Belg 2009, 64, 23-34, doi:10.1179/acb.2009.006.
  4. Mineharu, Y.; Koizumi, A.; Wada, Y.; Iso, H.; Watanabe, Y.; Date, C.; Yamamoto, A.; Kikuchi, S.; Inaba, Y.; Toyoshima, H., et al. Coffee, green tea, black tea and oolong tea consumption and risk of mortality from cardiovascular disease in Japanese men and women. J Epidemiol Community Health 2011, 65, 230-240, doi:10.1136/jech.2009.097311.

Round 2

Reviewer 2 Report

The Authors corrected the manuscriot according to the reviewers' comments. The paper is ready for publication.